# The suitability of native flowers as pollen sources for *Chrysoperla lucasina* (Neuroptera: Chrysopidae)

**Rafael Alcalá Herrera**[ID]**[1]\*, María Luisa Fernández Sierra[1], Francisca Ruano[2]**

**1** Department of Environmental Protection, Plant Protection Group, Estación Experimental del Zaidín (CSIC), Granada, Andalucía, Spain, **2** Department of Zoology, University of Granada, Granada, Andalucía, Spain

\* rafa.alcala@eez.csic.es

**Data Availability Statement:** Data has been uploaded to Digital CSIC with the following DOI and link: http://dx.doi.org/10.20350/digitalCSIC/12637.

## Abstract

Green lacewings (Neuroptera: Chrysopidae) are key biological control agents found in a broad range of crops. Given the importance of enhancing their presence and conservation, in this study, we aim to identify and to determine the relative importance of the pollen consumed by *Chrysoperla lucasina* (Lacroix, 1936) from 29 pollen types offered by 51 native plant species sown in an experimental farm in Villarrubia in the south of Spain. For the purposes of this study, *C. lucasina* specimens were captured in the late spring of 2016 and 2017. The pollen types and other components in the alimentary canal of *C. lucasina* were microscopically identified using the transparency method, which is a novel technique applied to green lacewings captured in the field. The results show that (i) *C. lucasina* feeds on over half of the pollen types offered by the sown plant species, with no differences in behaviour by sex or year; (ii) *Capsella bursa-pastoris* was the most frequently identified pollen type in the alimentary canal; (iii) the majority of pollen types identified correspond to sown native plant species and not to surrounding plant species; and that (iv) most of the adults studied also consumed honeydew. Our feeding study has important implications for the selection of plant mixtures for ground cover restoration and flower vegetation strips in Mediterranean agroecosystems, which complements our previous findings on how *C. lucasina* use native plant species as host and reproduction sites. The plant species *Capsella bursa-pastoris* and *Biscutella auriculata*, which are best suited to provide pollen, host and reproduction sites for *C. lucasina* in late spring, should consequently be included in the proposed plant mixtures for Mediterranean agroecosystems.

## Introduction

Agricultural intensification has increase global agricultural production, while also expanding agricultural land area and the use of external inputs such as pesticides, fertilizers and irrigation [1,2], as observed in Mediterranean vineyards, olive groves and fruit orchards [3–5]. However, these agricultural practices have had adverse environmental effects such as loss of natural habitats and biodiversity, leading to a decline in ecosystem services, particularly pest control and

**Funding:** This study was funded by a grant awarded to FR and Mercedes Campos from the Junta de Andalucía (project P12-AGR-1419) and a grant from Consejo Superior de Investigaciones Científicas (project 201840E055) to Mercedes Campos. The funders had no role in study design, data collection and analysis, decision to publish, or preparation of the manuscript.

**Competing interests:** The authors have declared that no competing interests exist.

crop pollination, as well as in soil structure and fertility [6,7]. It is vitally important to minimize negative environmental effects through effective ecological intensification strategies which require a good understanding of the relationship between land use and communities of organisms [8]. For example, the European Union has developed agri-environmental policies to reduce pesticides use and to promote biodiversity in agricultural landscapes through the protection and creation of semi-natural habitats (SNH) defined as habitats composed of non-crop plant species, both within and/or outside crop areas [9–11].

Conservation biological control, an alternative to the use of pesticides, seeks to attract and maintain natural enemies in crops by modifying and managing SNH [12]. Agricultural landscapes are characterized by a diverse range of SNH such as hedgerows, woodlands and forests, cover crops, herbaceous ungrazed habitats and grassy linear habitats [11,13]. These types of SNH, with their higher insect abundance and diversity than crops [14,15], have the potential to attract and support natural enemies of insect pests [16,17]. The relationship between plant species and insects is linked to reproduction and feeding [18]. Flowering plants attract insects through their floral resources, including pollen, nectar and honeydew, which are important at certain stages in the life of many natural enemies, as well as alternative prey and hosts, shelter and oviposition sites [19–22]. Although several grass and wildflower seed mixtures are currently available to farmers to increase arthropod abundance and diversity [15], commercial seed mixtures are less adapted to Mediterranean climates [23,24]. The use of native species could have a beneficial impact on ground cover while avoiding the disadvantages of non-native plant species [24–27]. Thus, knowledge of the ways in which different native plant species attract natural enemies and pollinators and of how their floral resources are used could contribute to the development of more effective mixtures and to the promotion of effective landscape management practices [28–31].

Lacewings, which belong to the functional group of natural enemies, are present in virtually all field crops around the world. They are some of the most studied predators of pests [32] and are widely used by integrated and organic pest management systems within the framework of classical biological control [33]. Green lacewings, which can be captured in herbaceous, scrub and tree strata [34–36], are dependent on the presence of SNH [17,37]. Species belonging to the complex *Chrysoperla carnea* (Stephens) are the most frequently identified and abundant green lacewings in cultivated areas [32], whose wide ecological and geographical distribution is due to their ecological versatility [35]. *C. carnea* (Stephens) sensu Henry [= *Chrysoperla affinis* (Stephens) sensu Thierry] and *Chrysoperla lucasina* (Lacroix, 1936), the first green lacewings to appear in field crops, are attracted to large patches of flowering plants [38], which are used as oviposition and feeding sites [20,28]. Green lacewing larvae prey on a wide range of small, soft-bodied insects, as well as on the eggs and small larvae of lepidopteran insects [39], while the adults, generally palyno-glycophagous, feed on honeydew, pollen and nectar [38,40–42]. Laboratory [43–45] and field [46,47] studies have concluded that proteinaceous food is necessary for lacewing egg production, while other studies have shown that flowering resources can influence life history parameters such as survival, reproduction and development time [45,48,49].

Analysis of pollen grains in the alimentary canal of palynophagous insects has revealed the habitats and plants visited, what they eat, as well as dispersal movements in and around agroecosystems [50]. Different methodologies have been used to extract alimentary canal contents from insects [51–54]. These include a process called acetolysis, by which an insect is destroyed by artificial acid digestion [50], rendering it impossible to determine where the pollen is located in the alimentary canal and thus to determine when the pollen were consumed. However, the transparency method can be used to identify and locate pollen in the insect's alimentary canal [55]. Analyses of pollen should be combined with plant inventories that provide the

set of available pollen resources for chrysopids [56,57]. Previous studies have reported that chrysopids, which consume pollen where and when available, mainly feed on herbaceous plant species belonging to the families Apiaceae, Asteraceae, Brassicaceae, Chenopodiaceae-Amaranthaceae, Fabaceae, Liliaceae, Plantaginaceae, as well as tree and shrub species from the families Ericaceae, Cistaceae, Oleaceae and Pinaceae [38,58]. Hence, knowledge about how insects in general, and more specifically chrysopids, use floral resources, particularly in SNH, is critical for successful habitat management strategies aimed at improving natural biological control [29,38,59,60].

The principal objective of this study is to identify and to determine the relative importance of the pollen consumed by *C. lucasina* adults by observing the alimentary canal of individuals collected from sown native plant species. Specifically, we were interested in:

1. whether *C. lucasina* is a generalist or specialist pollen feeder,

2. the principal sown native species exploited,

3. the importance of surrounding natural plants species in relation to the pollen consumed,

4. whether the pollen consumed by both sexes show a similar pattern,

5. and the possibility of identifying other components (fungal spore, honeydew) in the alimentary canal.

This should enable us to advance our knowledge of the feeding behaviour and ecology of *C. lucasina* and to suggest the most effective plant species to attract and conserve this key predator in Mediterranean agroecosystems.

## Material and methods

### Study area

The study was carried out on an experimental farm in the village of Villarrubia (37°49′49″N, 4°54′20″W) in the southern Spanish province of Cordoba. The farm is bordered by a commercial orange orchard, an olive orchard and various irrigated crops, as well as, to the south, by the Guadalquivir river, with its riverbank vegetation. In addition, *Pinus halepensis*, *Pinus pinaster* and *Pinus pinea* trees can be found in the Sierra Morena mountain range, five kilometres to the north of the farm [61]. No specific authorizations were required for the field study, which did not involve endangered or protected species.

In the experimental farm, in order to test the way in which chrysopid species exploited the plant resources, a set of 51 plant species were planted (S1 Table) which need to [62]: (i) be native angiosperms, (ii) be self-sowing winter annuals which do not compete for water with orchards during the summer season and (iii) have a bloom period prior to olive blooming in order to attract natural enemies to the agroecosystem. The potential use of these plants as ground cover and for seed production [25], as well as their attractiveness to on chrysopid species, were tested [28] in previous studies. The farm was tilled in late November 2015 and 2016, and the seeds were planted on consecutive days. The plots were irrigated once during germination and several times during plant development when necessary.

We sampled three 3x3 and 1x9 metre sampling areas (a total of 9 m$^2$) in 2016 and 2017, respectively, for every plant sown in each sampling event. In 2016, 40 plant species belonging to 13 families (S1 Fig and S2 Table) were randomly planted in three blocks of 40 plots. Each plant species was sown in three 3x3 m plots (a total of 120 samples). In 2017, although 35 plant species were sown in a single randomized plot configuration with different areas, we sampled only 20 plant species based on the chrysopid abundance results for 2016 sampling (S1 Fig and

S2 Table). The three samples (9 m$^2$) were located equidistantly from the centre of each plot to avoid border effect (60 samples in total).

Due to their poor development, five plant species sampled in 2016 (*Anarrhinum bellidifolium*, *Helianthemum ledifolium*, *Tuberaria guttata*, *Aegilops geniculata* and *Aegilops triuncialis*) and two species sampled in 2017 (*Medicago orbicularis* and *Medicago polymorpha*) were omitted from the analyses.

### Insect collection

Insects were collected only on emerged and well-developed blooming plant species (S2 Table). Three samples (9 m$^2$) per plant species were vacuumed twice in May 2016 and 2017 between 9 a.m and 2 p.m for 40 seconds using an InsectaZooka aspirator (BioQuip® Products Inc., Rancho Dominguez, CA, US). Samples were stored at -20˚C and cleaned in the laboratory. Chrysopid adults were identified under a stereomicroscope up to species level and were sexed according to the Iberian chrysopid key [63].

### Alimentary canal content analysis

The alimentary canal contents of all chrysopid adults collected were analysed by slightly modifying the transparency method described by Bello et al. [55]. Though previously used in studies of aquatic insect feeding and benthic fauna [64–67], this was the first time that this method was used to analyse chrysopid specimens collected from the field. The adult chrysopids were defrosted at room temperature, washed three times with distilled water and then vortexed to remove external pollen on the insect's surface. After removing the wings, legs and antennae, each individual was placed in a vial, covered with Hertwig´s solution and heated in an oven at 60˚C for a period of 24 hours. Two specimens, which were lost during the transparency manipulation process, were excluded from the pollen identification analysis. Subsequently, the specimens were mounted on a slide for microscopic examination under a DMI600B inverted microscope (Leica, Wetzlar, Germany) at 1000x magnification to identify the pollen. In addition, photomicrographs of the most frequently occurring pollen types were taken using a C-1 confocal microscope system (Nikon, Japan) at 1000x magnification to obtain a 3D picture of the shape of the pollen grain and to observe the autofluorescence of the exine according to the method described by Castro et al. [68]. Before pollen was identified, we divided the alimentary canal into five segments: I–head, II–thorax, III–initial abdomen, IV–medium abdomen and V–final abdomen. This was done to show the percentage of the alimentary canal occupied by the pollen in each segment, which was classified into three different categories: absent (0% grains), medium (under 50%) and high (50–100% occupation of the alimentary canal) at 40x magnification. The presence of pollen in each segment provided a rough approximation of the intensity of consumption, of when the pollen was consumed and of any differences between sexes with regard to the amount of pollen consumed.

Given that other components, such as nectar, honeydew, fungal spores, other fungi and/or arthropod exuviae, in the chrysopids alimentary canal have been previously reported [38,40,69,70], we visually checked their presence under an inverted microscope according to the method described by Lacey et al. [71]. Based on the findings of Villenave et al. [38,72], honeydew consumption was indirectly identified by the presence of fungal spores *Cladosporium sp*. and *Alternaria sp*.

### Pollen identification

We examined the pollen according to their morphological traits (polar and equatorial axes, shape, apertures and exine ornamentation), up to type (plant species and/or genus

assemblages), family, genus and/or species level. We identified the pollen using the method described by Valdés [73] and our reference pollen collection. This collection was created by growing the native plant species sown in the field under controlled greenhouse conditions: 25˚C, 50–60% humidity and a photoperiod of 16:8h (light:dark). The collection was deposited at the Department of Environmental Protection in the Estación Experimental del Zaidín, a Spanish Scientific Research Council (CSIC) centre in Granada.

## Statistical analyses

All analyses were carried out using the R software packages agricolae, FactoMineR, Factoextra and vegan [74–77]. As the data did not follow a normal distribution, the differences in the number of pollen types identified and the percentage of the alimentary canal containing pollen by sex were determined using the Kruskal-Wallis test with Bonferroni adjustment. Multiple correspondence analysis (MCA) was used to summarize and visualize the presence or absence of each pollen type in each adult chrysopid. MCA was applied to the different types of pollen recorded more than four times in the adult chrysopids examined in order to minimize the effect of pollen types occasionally consumed. As the sown native plant species differed slightly each year, we carried out MCA annually. A permutational multivariate analysis of variance (PERMANOVA), with Jaccard distance and 999 permutations, was then performed each year to determine whether pollen type composition differed significantly by sex in each year. Differences between sexes in relation to each fungal spore identified were also determined using the Fisher test in each year.

## Results

A total of 109 *C. lucasina* adults were collected (46 in 2016 and 63 in 2017), of which 71 were females (27 in 2016 and 44 in 2017) and 38 were males (19 in both 2016 and 2017). Overall, all adults examined were found to have pollen in their alimentary canal, with the highest percentages observed in the initial and medium abdomen (Fig 1). The percentage of alimentary canal containing pollen varied significantly: 39.2±2% in females as compared to 27.5±2% in males (Kruskal-Wallis test $\chi^2$ = 9.849, $p <$ 0.01). However, we observed very similar mean values for the number of pollen types by sex: 3.01±0.17 (n = 71 females) and 3.03±0.26 (n = 36 males) (Kruskal-Wallis test $\chi^2$ = 0.1, $p$ = 0.757).

The 51 native herbaceous plants sown between 2016 and 2017 in the experimental farm were pooled in relation to 29 pollen types (Table 1). *C. lucasina* consumed 17 of the 29 pollen types, with the following families being the most notable among the adults examined (Table 1): types *Anthemis arvensis*, *Calendula arvensis*, and *Crepis capillaris* on Asteraceae; types *Borago officinalis* and *Echium plantagineum* on Boraginaceae; types *Capsella bursa-pastoris* and *Biscutella auriculata* on Brassicaceae; and types *Silene latifolia*, *Silene vulgaris* and *Vaccaria hispanica* on Caryophyllaceae. Five pollen types from the following surrounding natural plants were also identified: Apiaceae, Ericaceae, *Trifolium arvense*, *Castanea sativa* and *Pinus pinea*.

Type *C. bursa-pastoris* was the pollen type most frequently identified in the alimentary canal (Table 1 and Fig 2). Despite some difficulty, we managed to differentiate between *B. auriculata* and *C. bursa-pastoris* pollen up to plant species according to grain pollen size (S2 Fig). We found that the most abundant pollen in the alimentary canal was from the species *C. bursa-pastoris*, which was thus identified in 86 of the 95 adults captured (35 specimens in 2016 and 51 in 2017), followed by *B. auriculata*, identified in 40 of the 95 adults captured (15 specimens in 2016 and 25 in 2017). We also recorded both plant species pollen (*C. bursa-pastoris* and *B. auriculata*) in 31 of the 95 captured adults (12 specimens in 2016 and 19 in 2017). These results are particularly noteworthy when we take into account that both plant species

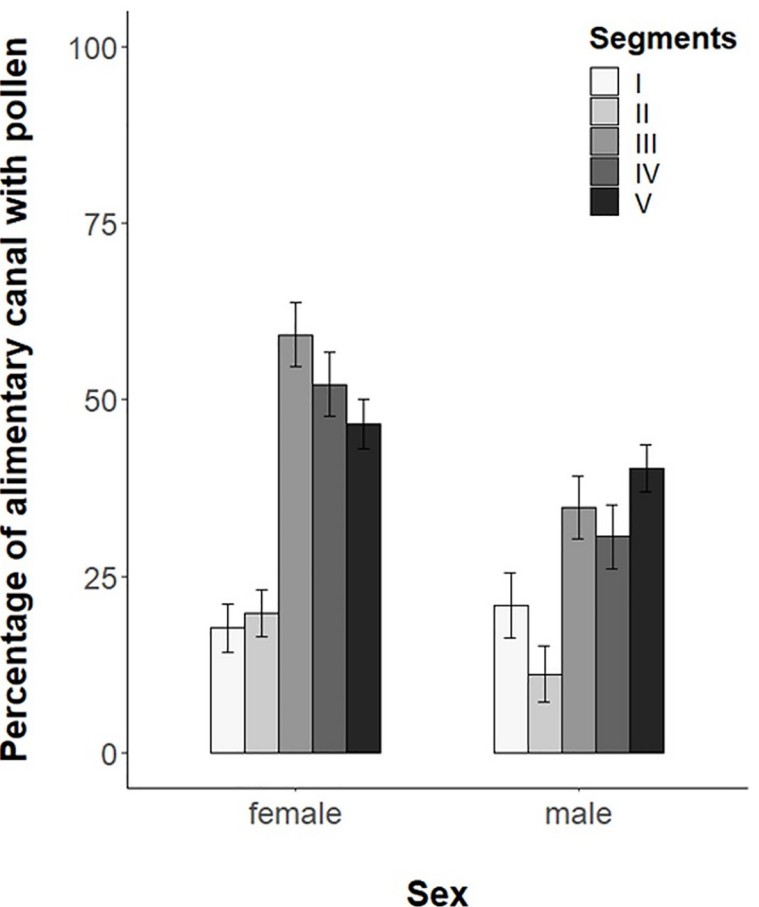

**Fig 1. Estimated percentage of alimentary canal containing pollen (mean ± SE) in each segment (I–head; II–thorax; III–initial abdomen; IV–medium abdomen and V–final abdomen) by sex.**

share some months during their bloom period (S2 Table); however, a slight difference in the phenology of both species was observed in 2016, with *C. bursa-pastoris* showing signs of senescence, while *B. auriculata* was blooming at the time of sampling. This explains why the plant species *C. bursa-pastoris* was not sown in the experimental farm in 2017.

We recorded similar pollen types in the alimentary canal of *C. lucasina* in both years studied; only seven pollen types, *C. arvensis*, *C. capillaris*, Ericaceae, *Mentha aquatica*, Rhamnaceae, *Salvia verbenaca* and *Scabiosa atropurpurea*, were absent from one of the sampling years, while twelve pollen types were not recorded in either of the sampling years (Table 1). The consumption of each pollen type by *C. lucasina* individuals showed different patterns of association, with no significant differences by sex in either year (Fig 3); PERMANOVA 2016 $R^2 = 0.038$, d.f. = 1, $p = 0.15$; 2017 $R^2 = 0.024$, d.f. = 1, $p = 0.173$. In 2016, the pollen consumed by *C. lucasina* were grouped in the following three pollen assemblages: *E. plantagineum*, *V. hispanica* and *S. vulgaris*; *C. bursa-pastoris* and *Festuca arundinacea*; and *P. pinea*, *A. arvensis* and *S. latifolia* isolates (Fig 3A). Meanwhile, in 2017, there were four groups: *C. capillaris*, *S. vulgaris*, Apiaceae, *F. arundinacea* and *V. hispanica*; *P. pinea*, *E. plantagineum* and *A. arvensis*; *Reseda luteola* and *S. verbenaca*; *S. latifolia*, *C. bursa-pastoris* and *T. arvensis* isolates (Fig 3B).

In addition to pollen, we identified fungal spores in the alimentary canal contents of 98 of the 107 *C. lucasina* adults examined. *Alternaria sp*. and *Cladosporium sp*. were the most

**Table 1. Presence (Y) or absence (N) of the different pollen types identified in *C. lucasina* adults collected in the experimental farm in 2016/2017.**

| Family | Pollen type | Pollen type identified in 2016/2017 |
|---|---|---|
| Apiaceae | *Type Apiaceae | Y(1)/Y(5) |
| | Type *Orlaya daucoides* | N/N |
| Asteraceae | Type *Anthemis arvensis* | Y(18)/Y(22) |
| | Type *Calendula arvensis* | Y(1)/N |
| | Type *Crepis capillaris* | N/Y(9) |
| Boraginaceae | Type *Borago officinalis* | Y(1)/Y(3) |
| | Type *Echium plantagineum* | Y(9)/Y(14) |
| Brassicaceae | Type *Capsella bursa-pastoris* | Y(38)/Y(57) |
| | Type *Raphanus raphanistrum* | N/N |
| Caprifoliaceae | Type *Scabiosa atropurpurea* | N/Y(1) |
| Caryophyllaceae | Type *Silene latifolia* | Y(4)/Y(13) |
| | Type *Silene vulgaris* | Y(8)/Y(18) |
| | Type *Vaccaria hispanica* | Y(9)/Y(16) |
| Cistaceae | Type *Helianthemum ledifolium* | N/N |
| | Type *Tuberaria guttata* | N/N |
| Ericaceae | *Type Ericaceae | Y(2)/N |
| Fabaceae | Type Fabaceae | N/N |
| | Type *Lotus creticus* | N/N |
| | Type *Trifolium arvense* | Y(2)/Y(4) |
| | Type *Trifolium repens* | N/N |
| Fagaceae | *Type *Castanea sativa* | Y(3)/Y(1) |
| Lamiaceae | Type *Lamium amplexicaule* | N/N |
| | Type *Mentha aquatica* | N/Y(3) |
| | Type *Salvia verbenaca* | N/Y(5) |
| Papaveraceae | Type *Papaver rhoeas* | Y(1)/Y(1) |
| Pinaceae | *Type *Pinus pinea* | Y(8)/Y(16) |
| Plantaginaceae | Type *Anarrhinum bellidifolium* | N/N |
| | Type Plantaginaceae | N/N |
| | Type *Plantago coronopus* | Y(1)/Y(1) |
| Poaceae | Type *Festuca arundinacea* | Y(6)/Y(9) |
| | Type Poaceae | N/N |
| Ranunculaceae | Type *Nigella damascena* | N/N |
| Resedaceae | Type *Reseda luteola* | Y(2)/Y(10) |
| Rhamnaceae | *Type Rhamnaceae | N/Y(1) |

The number of chrysopid specimens in which each type of pollen was identified in each year is in parenthesis.

*Pollen type identified from surrounding vegetation.

frequently identified spores, with no significant differences by sex observed in either year (2016 Fisher tests: *Alternaria sp.* $p = 0.640$; *Cladosporium sp.* $p = 0.359$; *Spore1 sp.* $p = 0.355$; *Spore2 sp.* $p = 1$, and 2017 Fisher tests: *Alternaria sp.* $p = 0.224$; *Cladosporium sp.* $p = 0.780$; *Spore1 sp.* $p = 0.785$; *Spore2 sp.* $p = 1$) (Fig 4).

## Discussion

In this study, the alimentary canal contents of *C. lucasina* provided great insight into their late spring feeding behaviour. The transparency method also enabled us to show that *C. lucasina* had mainly fed on the plant species sown and to investigate their previous visits to plants.

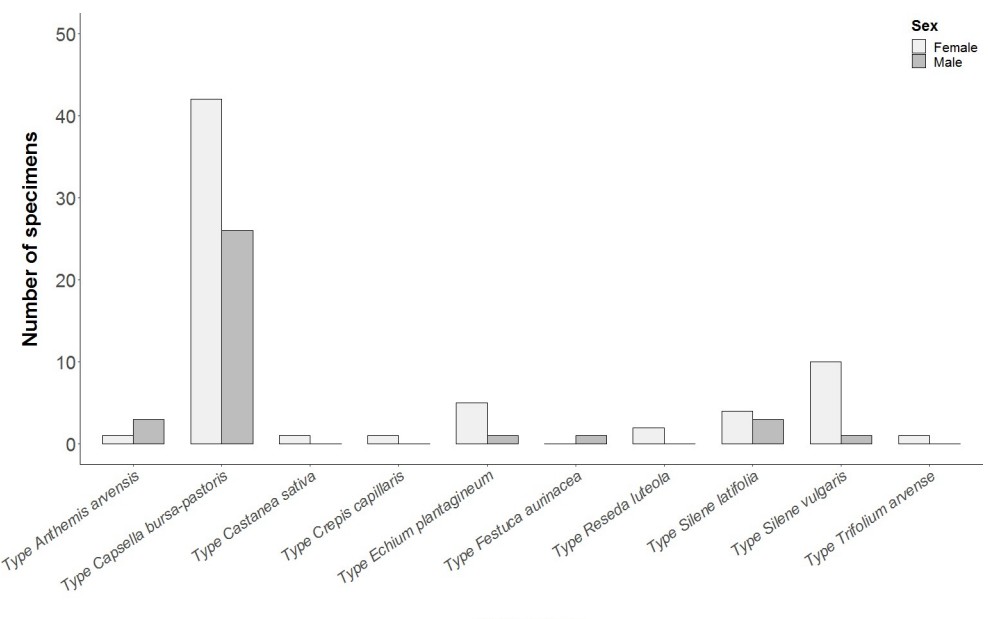

**Fig 2. Pollen type most frequently identified in the alimentary canal of *C. lucasina* adults by sex.**

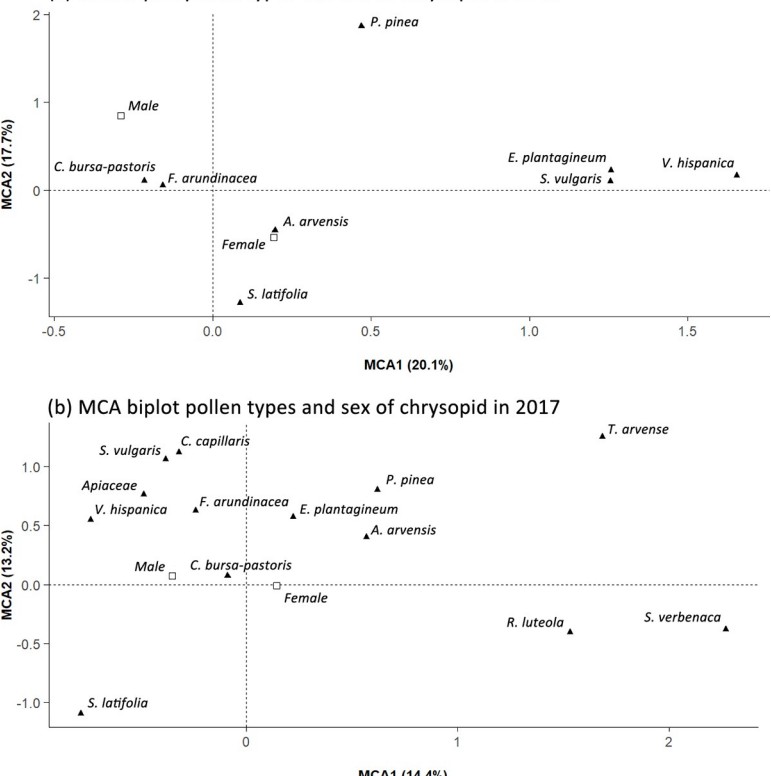

**Fig 3.** MCA biplot shows the association between the pollen types identified (▲) and sex of chrysopids (□) in 2016 (a) and 2017 (b).

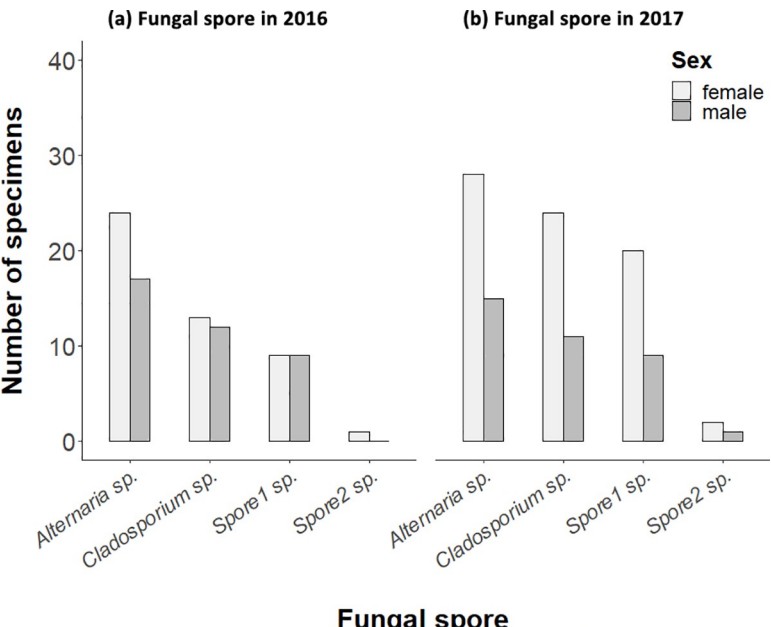

**Fig 4.** Fungal spore identified on chrysopid adults by sex in 2016 (a) and 2017 (b).

We show that 17 of the 29 pollen types on the 51 native plants sown were consumed by *C. lucasina*, which have a rich polyphagous diet without any differences observed by sex. Furthermore, multiple correspondence analyses found no stable pattern of association in either year among the pollen types consumed, thus confirming that *C. lucasina* is not a specialist pollen feeder. We also observed that the principal pollen types in the alimentary canal of *C. lucasina* were type *C. bursa-pastoris* (including *B. auriculata* and *C. bursa-pastoris* plant species) belonging to the Brassicaceae family, as well as type *A. arvensis* from the Asteraceae family which, according to previous studies, are important sources of pollen for *C. carnea s.l.* especially in spring [38,58]. Most of the pollen types identified have been reported in previous studies of the feeding patterns of chrysopids, whose feeding habits were shown to be opportunistic and eclectic, although, in some cases, their diet was found to specialize in particular pollen types [29,36,38,57,58,78]. These feeding differences in chrysopids could be explained by the variety of plant species in certain habitats and their availability for chrysopids, but might also be due to different chrysopid species, with probably slightly different feeding behaviours, present in the *C. carnea* complex [29,57,58]. It might be necessary to carry out further research into the 12 pollen types in our study that did not appear in the chrysopid alimentary canal in order to determine feeding preferences when type *C. bursa-pastoris* is unavailable in the landscape.

*C. lucasina* emerge in spring and is one of the most frequently identified and abundant chrysopid species during the blooming stage of herbaceous vegetation [36,79]. Green lacewing populations have different patterns of activity and levels abundance depending on the species and stage of **development** and habitat, as observed in vineyards, with fewer captures observed during the month of May [17]. Thus, the provision of sown plants which bloom when chrysopid populations decreasing could contribute to increasing predator/prey ratios at key moments in order to improve biological control.

In a previous study, we selected 42 native plant species to evaluate their attractiveness to chrysopids based on chrysopid captures from sown plant species and on their use as

reproduction sites [28]. By linking these previously published findings to the current results, we observed that one of the most attractive plant species for chrysopids is *B. auriculata*, whose pollen is the second more abundant in the alimentary canal of *C. lucasina*. Immature and adult stage *C. lucasina* were found on *B. auriculata*, which was visited for feeding, reproductive and refuge purposes. On the one hand, the lack of captured adult or immature stage chrysopids in 2016 from *C. bursa-pastoris* can be explained by signs of plant senescence at the moment of sampling. Although *C. bursa-pastoris* was not on the sown plant list for 2017 [28], its pollen was consumed by 81% of the chrysopid adults examined as compared to 40% from *B. auriculata* in that year. We therefore concluded that *C. bursa-pastoris* selectively attracts *C. lucasina*. We also observed that the plant species such as *Nigella damascena*, *Moricandia moricandioides* and *Silene arvensis* [28], most visited by chrysopids are not used as pollen sources by *C. lucasina*. However, other plant species, such as *B. officinalis*, *E. plantagineum* and *P. rhoeas* were abundantly consumed in certain chrysopid specimens despite the presence of bees and bumblebees in these plant species at the time of sampling. The present study demonstrates that, given the feeding preferences of *C. lucasina*, the plant species *C. bursa-pastoris* and *B. auriculata* should be included in conservation biological control programs.

Our findings could also facilitate the selection of suitable plant species in the context of a probable shift from woody pollen sources in spring to herbaceous pollen sources in summer according to the study by Bertrand et al. of *C. carnea* [29] given that the blooming period of the sown plant species in our study coincides with *C. lucasina* activity in Mediterranean agroecosystems [80]. Villa et al. [58] found that *C. carnea s.l.* mainly feeds on *Olea europaea* (woody strata) in all seasons and to a lesser extent on herbaceous plants in spring and summer. As suggested by Villenave et al. [38], this highlights the importance of providing plant species with different blooming periods that correspond to seasonal chrysopid flight activity and, as recommended by Bertrand et al. [29] that of increasing plant and vegetation diversity to attract chrysopid populations to agroecosystems. Despite the availability of several flowering plant species over time, Villenave et al. [38] have shown that *C. lucasina* is attracted to large patches of flowering plant species. In our study, although the pollen type *C. bursa-pastoris* was consumed, the plant species was sown in small 9 m$^2$ patches and even in surrounding non-sown plants.

We also found that the alimentary canal of *C. lucasina* females contained significantly more pollen than that of males. The number of pollen grains observed by Villenave et al. [36] in the diverticulum, an alimentary canal structure of adult chrysopids where the pollen is stored before being digested, is similar to that reported in our study. In our view, the differences between sexes could be related to the higher feeding requirements of female green lacewings as compared to those of males. A previous study found that protein titre levels significantly affect the reproductive physiology and fecundity of *C. carnea* [81]. Females reach protein titre levels during egg laying on roughly the fourth or fifth day following emergence, while males reach this point at the end of the second day [81]. Thus, the proteinaceous food provided by flowers is essential for the reproduction of adult lacewings [40,43–47]. However, the floral architecture and nectaries need to be accessible to ensure that these food resources can be reached [49]. Furthermore, as found for bees by Liolios et al. [82], the nutritional requirements of chrysopids are probably satisfied by a small number of abundant plants with accessible pollen present in the landscape and blooming for long periods of time in a manner unrelated to pollen protein content. This could be related to the attraction of green lacewings to large patches of flowering plants [38], the reported consumption by *C. carnea s.l.* of mostly *O. europaea* pollen in olive agroecosystems [58] or our results regarding its feeding on *C. bursa-pastoris* and *B. auriculata*. Nevertheless, Liolios et al. [82] have observed that the protein content of *O. europaea* pollen is lower than that of other common plant species surrounding olive orchards, such as *Reseda sp.*, *Papaver rhoeas*, *Trifolium sp.* and *S. verbenaca*, which were also sown for our study.

Given the high capacity of chrysopid movements [83–85], we also identified mixtures of entomophilous and/or anemophilous pollen types from surrounding trees and herbaceous vegetation such as Ericaceae, *C. sativa*, *P. pinea*, *Foeniculum vulgare* and Rhamnaceae which were also reported as chrysopid resources in previous studies [36,58]. The presence of these pollen types was low in the *C. lucasina* alimentary canal as compared to the pollen types offered by the sown plant species. Anemophilous plant species, such as type *P. pinea*, whose pollen may have been consumed on vegetation surfaces, were at the blooming stage in our study area and at the time of sampling [86]. Furthermore, as *C. lucasina* had already been known to use *P. halepensis* for reproductive purposes [80], its use of this tree stratum for feeding and refuge cannot be ruled out.

With respect to honeydew, Villenave et al. [38,72] managed to indirectly identify its consumption through the presence of fungal spores, arthropod exuviae and pollen from anemophilous plant species in the alimentary canal of chrysopids. We found that most chrysopid adults have fungal spores in their alimentary canal, while previous studies have reported that *Alternaria sp.* and *Cladosporium sp.* fungal spores reaching peak abundance during the months of May and June in our study area [87,88]. We therefore concluded that a large proportion of *C. lucasina* adults feed on honeydew in late spring given the presence of fungal spores and anemophilous pollen in the alimentary canal of chrysopids.

Finally, in our view, although flower visitation studies are an important aid to selecting suitable plant species as refuges and reproduction sites, they need to be supplemented by flower foraging analyses. Only then can a full picture be obtained of how chrysopids exploited plant species and of how to guarantee their presence in the required habitat and time.

## Conclusions

This study shows that the *C. lucasina* captured feed mainly on pollen from sown native species (>77%), with no differences observed by sex or year, and that *C. bursa-pastoris* was the most abundant pollen type consumed by *C. lucasina*. The identification of anemophilous pollen and fungal spores in its alimentary canal, confirms that *C. lucasina* feeds on honeydew on vegetation surfaces. Our recommendation at the conclusion of this study is to include *C. bursa-pastoris* and *B. auriculata* in the list of the most attractive plant species for *C. lucasina*. These native plant species are associated with the most frequently consumed pollen types identified in its alimentary canal, are able to attract and maintain *C. lucasina* populations and also promote a more heterogeneous landscape in Mediterranean agroecosystems. Finally, analysis of floral visits by chrysopids should be complemented with feeding studies in order to provide a more complete picture of plant resources use by chrysopids.

## Supporting information

**S1 Fig.** Field layout with sown plant species distribution in the experimental farm in 2016 (A) and 2017 (B).
(PDF)

**S2 Fig.** *Biscutella auriculata* (A and B) and *Capsella bursa-pastoris* (C and D) flowers; photomicrographs of *B. auriculata* (E) and *C. bursa-pastoris* (F) pollen grains taken by a confocal microscope at 1000x magnification. Both images (E and F) show exine autofluorescence after merging multiple optical sections; microscopic images of *B. auriculata* (G) and *C. bursa-pastoris* (H) pollen grains at 1000x magnification.
(PDF)

**S1 Table. List of plant species used in the study.**
(PDF)

**S2 Table. Complete list of plant families, pollen types, plant species, year sown, year sampled, chrysopid collected, bloom period, plant type and lacewing feeding studies references.** Y–Yes, N–No.
(PDF)

# Acknowledgments

We wish to thank Dr. Mercedes Campos for assistance obtaining funding, laboratory support and encouraging us to publish this study, Luis Plaza and Joaquin Chocano for assistance in the field, Dr. Manolo Tierno for transparency methodology training and Dr. Cándido Galvez for resolving pollen identification issues. Finally, we would like to thank Michael O'Shea for proofreading the manuscript.

# Author Contributions

**Conceptualization:** Rafael Alcalá Herrera, Francisca Ruano.

**Data curation:** Rafael Alcalá Herrera.

**Formal analysis:** Rafael Alcalá Herrera, Francisca Ruano.

**Funding acquisition:** Francisca Ruano.

**Investigation:** Rafael Alcalá Herrera, María Luisa Fernández Sierra.

**Methodology:** Rafael Alcalá Herrera.

**Resources:** Rafael Alcalá Herrera, María Luisa Fernández Sierra.

**Supervision:** Rafael Alcalá Herrera, Francisca Ruano.

**Validation:** Rafael Alcalá Herrera, Francisca Ruano.

**Visualization:** Rafael Alcalá Herrera, María Luisa Fernández Sierra.

**Writing – original draft:** Rafael Alcalá Herrera.

**Writing – review & editing:** Rafael Alcalá Herrera, María Luisa Fernández Sierra, Francisca Ruano.

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
