## [Decision Letter · Decision Letter 0]

18 May 2020

PONE-D-20-10818

The suitability of native flowers as pollen source for Chrysoperla lucasina (Neuroptera: Chrysopidae)

PLOS ONE

Dear Dr. Alcalá Herrera,

Thank you for submitting your manuscript to PLOS ONE. After careful consideration, we feel that it has merit but does not fully meet PLOS ONE’s publication criteria as it currently stands. Therefore, we invite you to submit a revised version of the manuscript that addresses the points raised during the review process.

I agree with both reviewers that the paper needs a lot more detail throughout. In particular, the Introduction would benefit from more detail on the importance and broader relevance of this study, and the Methods need much more detail about the collections, the plants used in the epxeriment, and justification for why particular methods were used. The authors should also carefully edit the language used throughout the manuscript to avoid misinterpretations; this is an observational study, behavioural factors (i.e. feeding preferences) were not tested here.

We would appreciate receiving your revised manuscript by Jul 02 2020 11:59PM. To enhance the reproducibility of your results, we recommend that if applicable you deposit your laboratory protocols in protocols.io, where a protocol can be assigned its own identifier (DOI) such that it can be cited independently in the future. For instructions see: http://journals.plos.org/plosone/s/submission-guidelines#loc-laboratory-protocols

We look forward to receiving your revised manuscript.

Kind regards,

Manu E. Saunders

Academic Editor

PLOS ONE

Journal Requirements:

Additional Editor Comments (if provided):

Reviewers' comments:

Reviewer's Responses to Questions

**Comments to the Author**

1. Is the manuscript technically sound, and do the data support the conclusions?

Reviewer #1: Partly

Reviewer #2: Yes

2. Has the statistical analysis been performed appropriately and rigorously? 

Reviewer #1: N/A

Reviewer #2: Yes

3. Have the authors made all data underlying the findings in their manuscript fully available?

Reviewer #1: No

Reviewer #2: Yes

4. Is the manuscript presented in an intelligible fashion and written in standard English?

Reviewer #1: Yes

Reviewer #2: Yes

5. Review Comments to the Author

Reviewer #1: In the current manuscript, the authors investigated the feeding behaviour of the chrysopid species Chrysoperla lucasina for different types of pollen obtained from 51 native plant species from an experimental farm in Spain. The authors found that this chrysopid species fed on various plants, consuming 17 out of the 29 pollen types available. Some methods and results presented by the authors are not reliable and some details are missing. In the method section, the sampling area and the plant species used by the authors are not mentioned. They also don’t mention the number of plants used (replicates) and how plants were distributed in the farm. Something that also needs be noted is the number of insects that the authors sampled. In 2016, they sampled 45 individuals and in 2017, they sampled 62. If the authors sown 51 plant species in the farm, sampling only 45 chrysopid is not enough. Furthermore, in several parts of the manuscript, the authors claimed they have tested the feeding preference of the chrysopid, which is also misleading. They did not perform any preference test. Table and figures need to be improved (need to be more informative and should align with the goals of the manuscript). Therefore, due to all these issues, I am rejecting the publication of this manuscript.

I recommend the authors to be focused on their goals and on the scientific contribution of the manuscript while re-writing the text and I hope my comments bellow help with that.

COMMENTS TO THE AUTHORS:

Abstract:

Line 14: Species name should be writing without abbreviation when cited for the first time.

Line 24: Can you be sure that the insects consumed honeydew?

Introduction:

The manuscript is related to native plants of a specific region in South of Spain, but in the introduction, the authors don’t provide info for questions such as: Why is it important to study this region? Why is it important to study green lacewings (more specific, C. lucasina) in this area? What is the problem you can try to solve? I suggest to author include some info about this. Write more clearly the aims of the study and follow it through the text.

Line 36: I suggest:"…to minimize negative effects in the environment through ..."

Line 40: What are semi-natural habitats (SNHs)? Maybe include just one sentence to explain it better, because right after that, you talk about “low SNHs.

Line 45: I suggest “Flowering plants attracts insects through its floral resources, ...”

Line 47: When the author says entomophagous insects, I suggest to add some explanation or synonym such as predator, carnivorous insects or natural enemies.

Line 51: The conclusion of the sentence (“depending on the target arthropod”) seems loose.

Line 54: Is it correct to say “complex Chrysoperla carnea (Stephens)”?

Line 64: I suggest you to remove this sentence (C. carnea require a floral architecture sufficiently large for their thorax and mouthpart to reach the pollen and well-exposed nectaries to gain access to nectar [27-29].).

Line 76: “species-richness inventories” are referring to plant species or insect species? This paragraph needs a conclusion sentence.

Material and Methods

I suggest you to split the topic “Study area” in two: “Study area” and “Insect collection” (or something like that). Again, why is it important to study green lacewings in this area? What is the problem to be solved? In the topic “Insect collection” write more details about the sampling effort, how the sampling was made….

Include details about the native species that were sown and why they were selected?

Results

The results are confused, as well as the Table 1. The figure 1 have two variables mixed: pollen type and plant species, is it correct? Line 179 to 185: In the first part, you used “first, second and third” to refer to groups. Then, you use numbers “1, 2, 3 and 4”. I suggest you to choose only one way to refer to groups.

Discussion

In general, I suggest you to explore more your results.

Lines 224 to 232: this paragraph could be in the result section. It is simple and clear.

Conclusion

Line 272: “We specifically suggest that the sources of pollen include the following native plant species … which are the most frequently consumed by C. lucasina.” Do you suggest the use of these species everywhere or only at your study region? Be more cautious with your conclusions and keep the focus.

Reviewer #2: I have reviewed the article entitled “The suitability of native flowers as pollen source for Chrysoperla lucasina (Neuroptera: Chrysopidae)”. This is an interesting article addressing an important topic for conservation biological control of pests. Despite that there are other similar studies published, I believe it is important that this study is focused on C. lucasina, which was previously neglected or confounded with C. carnea, so any advance on knowing the feeding behavior and ecology of the species from the carnea complex will help us understand more about their ecology, the differences between and, from an applied point of view, how to enhance their abundance in cultivated fields.

I congratulate the authors for their work, given that identifying the pollen grains is complicated, and the use of the transparency method seems like a good option, easier than the acetolysis process that is more commonly used. Nevertheless, I think that the article would benefit from some changes before being published. Below, I provide my main concerns and some minor suggestions to the authors.

Major comments:

-The introduction is good and deals with relevant topics for the study system. However, at some points, it can be improved. The first paragraph would benefit from more connectivity between the sentences, as some of them are a bit disconnected, leading to the ecological intensification of agriculture. At the end of the second paragraph, the authors jump directly to chrysopids when they were talking about biological control and natural enemies in general. I believe that the link with lacewings can be included directly in the next paragraph or that the last sentence should be slightly modified to incorporate them in this general context. The third paragraph is a bit disorganized and would benefit from a better structure when talking about different aspects of lacewings biology and feeding preferences.

-The objectives of the article focus on pollen feeding preferences but do not mention the differences between sexes and the importance of honeydew for lacewings. However, both topics are addressed in the results and specific figures are shown. I think both topics are relevant, especially honeydew consumption, so the authors could include these as specific objectives or questions. During the introduction and methods, the fact that some lacewings feed on honeydew in briefly mentioned, but only at the end of the discussion the authors explain that this can be identified by the presence of fungal spores. I think this should be clarified before, in the introduction or methods section.

-The plant species sown are listed in table 1 in the results section and, in methods, this is only briefly mentioned (L. 98). However, the selection and sowing of the species is part of their methods, so I think that the reasons for their choice can be included before. The authors could refer to table 1 in methods, but explain if the plant species included flowers with different traits, flowering periods, previous reports on lacewing feeding, etc.

-In the multivariate analyses of pollen consumption, the authors refer to the overlap between diets in results, but this could be explored explicitly. A specific test such as an ANOSIM or similar could be done to determine if the differences between sexes on the diet composition were significantly different or not.

-As with the objectives section, I think the results section could be organized to refer to different specific objectives. The section has some order, but I have the impression that results from different questions are mixed and can be organized better. In my opinion, different parts that can be separated better are the description of the diet including the presence of surrounding plants, feeding preferences (including the multivariate analyses), differences by sex, presence of spores, and honeydew.

-In the discussion (l. 226-231) and conclusions (l. 272-275) the mention of a lot of plant species names becomes too repetitive. Furthermore, the temporal variation in the diet of lacewings should be discussed more, as the authors only sampled in May, and the diet changes through time. Regarding this, a recent and very relevant paper (Bertrand et al. 2019, J Appl Ecol, 56(11), 2431-2442) should be also mentioned.

-The article is, in general, well written, but would benefit from the language edit by a native English speaker. The parts where corrections are necessary are just a few, but it will improve the text flow. Some of these details are highlighted below in my minor comments.

Minor comments:

-The title should say "as pollen sources" or "as a pollen source" instead.

-L. 23: the sentence from point (iii) of the results section of the abstract is not so clearly written. I believe it would be more correct to say that the majority of the pollen types identified correspond to the sown native plants, and no that the pollen was found on the plant species.

-Throughout the manuscript, the authors refer to semi-natural habitats as SNH or SNHs, not always because they refer to singular or plural mentions of these habitats. I believe that SNH is widely used in the literature with no need to add the ‘s’ in the end and the same criteria should be applied across the text.

-L. 52: the family of natural enemies sounds strange, as it can be confounded with taxonomical families. The group or functional group might be more appropriate.

-L. 83: The sentence would be improved by adding “to identify which/the sown native species”.

-L. 58: Many references are cited like Villenave, Deutsch in this line. Please check the journal guidelines as I believe that in these cases only the first author should be mentioned before the number of the reference.

-L. 90: the section describes the study area but also lacewing samplings. The title could then be modified to make this clear?

-L. 108: is “peeling off” the best term to describe this? Taking off wings, legs, and antennae might be better.

-L. 116: the division of the alimentary canal in sections is interesting, but it is not clear if the authors did it after identifying the pollen or if the identification was done without contemplating the location of the pollen grains and the division was done just to show the proportions of the locations. A better explanation of the reasons can be helpful here.

-L. 137-138: instead of “in addition to”, the authors might refer to “using the packages”.

-L. 145: I would not say that including the species that were consumed only a few times would cause disturbances in the analysis, but they could be influential. Thus, perhaps “to minimize the influence of species that were only consumed occasionally” or something similar could be a better expression.

-Table 1 could be organized in a different way to show the relative importance of plant families. Instead of organizing it alphabetically by pollen type names, can the authors organize it alphabetically by plant family and then by pollen type within each family?

-L 271-272: pollen sources and resources could be changed to pollen resources or plants providing pollen resources.

-Fig. 2 would benefit from titles on each panel for better identification of what it is shown.

6. PLOS authors have the option to publish the peer review history of their article (what does this mean?). If published, this will include your full peer review and any attached files.

Reviewer #1: No

Reviewer #2: Yes: Ezequiel Gonzalez

---

## [Author Response · Author response to Decision Letter 0]

3 Jul 2020

We have answered the reviewers’ comments in a separate file titled "Response to Reviewers".

---

## [Decision Letter · Decision Letter 1]

13 Aug 2020

PONE-D-20-10818R1

The suitability of native flowers as pollen sources for Chrysoperla lucasina (Neuroptera: Chrysopidae)

PLOS ONE

Dear Dr. Alcalá Herrera,

Thank you for submitting your manuscript to PLOS ONE. After careful consideration, we feel that it has merit but does not fully meet PLOS ONE’s publication criteria as it currently stands. Therefore, we invite you to submit a revised version of the manuscript that addresses the points raised during the review process.

Please address the comments from Reviewer 1, and especially ensure the Introduction and Discussion sections are clear and well-organised.

We look forward to receiving your revised manuscript.

Kind regards,

Manu E. Saunders

Academic Editor

PLOS ONE

Reviewers' comments:

Reviewer's Responses to Questions

**Comments to the Author**

1. If the authors have adequately addressed your comments raised in a previous round of review and you feel that this manuscript is now acceptable for publication, you may indicate that here to bypass the “Comments to the Author” section, enter your conflict of interest statement in the “Confidential to Editor” section, and submit your "Accept" recommendation.

Reviewer #2: (No Response)

Reviewer #3: All comments have been addressed

2. Is the manuscript technically sound, and do the data support the conclusions?

Reviewer #2: Yes

Reviewer #3: Yes

3. Has the statistical analysis been performed appropriately and rigorously? 

Reviewer #2: Yes

Reviewer #3: Yes

4. Have the authors made all data underlying the findings in their manuscript fully available?

Reviewer #2: Yes

Reviewer #3: Yes

5. Is the manuscript presented in an intelligible fashion and written in standard English?

Reviewer #2: Yes

Reviewer #3: Yes

6. Review Comments to the Author

Reviewer #2: In the revised version of the manuscript “The suitability of native flowers as pollen source for Chrysoperla lucasina (Neuroptera: Chrysopidae)”, the authors have replied to the comments made by the two reviewers during the first submission. I thank the authors for their work on the revision. They have placed more emphasis on the importance of the study for the Mediterranean region, which helps the reader to understand why this research was relevant. English writing is also improved, but still, some changes are necessary, especially in parts of the discussion. The manuscript is clearer now and the objectives are clearly defined. Nevertheless, I think that the discussion needs more structure and some important topics are omitted. Plus, I found many minor changes that would improve the text. Therefore, I am recommending another major revision.

Major comments:

-Despite that their previous article (Alcalá Herrera et al. 2019, Biol. Control) is cited in the text, I feel that more links to this paper are necessary, as both works are clearly part of the same project and are highly connected. Particularly, in the discussion, it could be mentioned if the plants that were frequently visited by lacewings in their previous article are the same that were found to be consumed here. By looking at the highlights and abstract, it seems that not all species are shared, at least within the most important. What are the implications of this? Could the recommendations of sown plant species based only on flower visitation be inaccurate?

-In general, the discussion structure seems unorganized. The first paragraph is quite long and mixes their main findings with the discussion about other aspects of lacewings’ identification, temporal dynamics, and movement. I suggest to keep the first paragraph short and with a clear mention of the contribution of their study and discuss the other topics in other parts. Furthermore, on lines 267-269 the question of whether C. lucasina is a generalist or specialist pollen feeder appears, and this was not mentioned before, at least explicitly. The specific objectives could introduce this question more clearly, perhaps. The paragraph on the competition with other insects such as pollinators on lines 306-313 is disconnected from the rest of the article, as this was not mentioned before, so some changes to improve the connection are needed. Finally, the result of significant differences between sexes in the percentage of alimentary canal with pollen are not discussed in detail, and this could be important due to the implications of flower resources on reproduction (which is mentioned in lines 295-305, where this difference could be discussed).

Minor comments:

-L 13-16: I think that the revised description of the objective in the abstract can still be improved. The authors did not only want to identify the consumed pollen but also to quantify their relative importance. I agree with reviewer #1 that there are not measuring preferences explicitly, but at least the relative consumption of the different species was analysed and it would be good to highlight it here.

-L 19-20: Not clear. Did the authors mean that it feeds on more than half of the pollen types offered by the sown plants?

-L 32-33: Agricultural intensification is also very evident in annual crops, perhaps even more than on vineyards and orchards. Maybe changing “especially” for something like “as observed in” can retain the same meaning without emphasizing on the difference between different production systems.

-L 38: A better connection with the sentence about the European agri-environment schemes would improve the end of the paragraph, such as “For example, the European…”.

-L 40: delete the comma after “defined as habitats”.

-L 73-74; the expression “are subject to” is not ideal for this sentence. I would say that flowering resources can influence or affect these life-history parameters rather than being subject to them.

-L 82-83: this sentence is not clear, particularly the species-rich flowering plants part. Analyses of pollen should be combined with plant inventories or samplings that provide the set of available pollen resources.

-L 92-93: I would replace “To achieve this goal, we needed to determine” for a shorter expression such as “Specifically, we were interested in”, or something similar that makes a better connection between the general and specific objectives because those four objectives are specific questions asked by the authors.

-L 104-105: I understand that this information is necessary for the methods section according to the journal. But I believe that the beginning of the section is not the best option. Perhaps at the end of the paragraph, it would be better?

-L 110: can be found is repeated twice in this sentence.

-L111: plant species for what? This is the first time the authors mention the sowing of the flowering plants in methods, so the paragraph should start by saying that in this experimental farm, a set of X plant species were planted, following those criteria.

-L 119: squares should be replaced by plots or something alike, as the 1x9 m sampling areas are not squares. This is also true for the rest of the paragraph and in other parts of the methods section.

-L 122: some plots were not well developed? It should be 120.

-L127-130: the writing on this short gives the idea that only those five species were sampled in 2016 and those two in 2017, because of the addition of “the” before the species number. I suggest to change it for something like “Due to their poor development, five species sampled in 2016 (species names) and two species sampled in 2017 (species names) were omitted from the analyses”.

-L 191: Please correct, you did not check for differences among years, but among sexes on each year (at least that is shown in results).

-L 200: the “also” gives the impression that no differences were found for the percentage of the alimentary canal in the previous sentence, but you did found differences.

-L 209: remove the comma after C. lucasina.

-L 224: confusing redaction, rephrase “the pollen types most recorded among the ten most frequently identified”.

-L 239-243: what information is provided by these groups? This is not discussed later in the article and it is probably linked with some individuals sharing some pollen types, but if it doesn’t represent something relevant, I would remove this description of the groups from the results.

-L 250-253: again, no differences between years were tested, rephrase. Also, the start of the parenthesis is missing and those are several Fisher tests, in the plural.

-L 288-289: this sentence is not clear.

-L 314-324: were these anemophilous plants represented by only a few pollen grains per individual lacewings or by many? In previous articles and my experience, C. sativa could be used as a direct pollen source because many grains can be observed sometimes. But Pinus and other coniferous are sometimes represented by very few grains and this could be a clear indication that lacewings consumed these species on the surface of other plants or with honeydew.

Reviewer #3: The authors appear to have carefully addressed all queries and comments raised in review and provision of a marked copy enabled efficient assessment of resubmission

7. PLOS authors have the option to publish the peer review history of their article (what does this mean?). If published, this will include your full peer review and any attached files.

Reviewer #2: No

Reviewer #3: **Yes: **Linda J. Thomson

---

## [Author Response · Author response to Decision Letter 1]

10 Sep 2020

Please see Response to Reviewers letter at the end of this PDF

---

## [Editor Report · Decision Letter 2]

15 Sep 2020

The suitability of native flowers as pollen sources for Chrysoperla lucasina (Neuroptera: Chrysopidae)

PONE-D-20-10818R2

Dear Dr. Alcalá Herrera,

We’re pleased to inform you that your manuscript has been judged scientifically suitable for publication and will be formally accepted for publication once it meets all outstanding technical requirements.

Kind regards,

Manu E. Saunders

Academic Editor

PLOS ONE
---

## [Editor Report · Acceptance letter]

14 Oct 2020

PONE-D-20-10818R2 

The suitability of native flowers as pollen sources for *Chrysoperla lucasina* (Neuroptera: Chrysopidae) 

Dear Dr. Alcalá Herrera:

I'm pleased to inform you that your manuscript has been deemed suitable for publication in PLOS ONE. Congratulations! Your manuscript is now with our production department. 

Kind regards, 

on behalf of

Dr. Manu E. Saunders 

Academic Editor

PLOS ONE